# Exercise Capacity in Very Low Birth Weight Adults: A Systematic Review and Meta-Analysis

**DOI:** 10.3390/children10081427

**Published:** 2023-08-21

**Authors:** Grace Poole, Christopher Harris, Anne Greenough

**Affiliations:** 1Neonatal Intensive Care Centre, King’s College Hospital NHS Foundation Trust, London SE5 9RS, UK; grace.poole5@nhs.net (G.P.); christopher.harris@kcl.ac.uk (C.H.); 2Department of Women and Children’s Health, Faculty of Life Sciences and Medicine, King’s College London, London SE5 9RS, UK

**Keywords:** very low birth weight, VLBW, neonatal intensive care, VO_2_ max, exercise capacity, exercise tolerance, physical activity

## Abstract

There is an association between very low birth weight (VLBW) and cardiovascular morbidity and mortality in adulthood. Aerobic fitness, measured as the maximal oxygen consumption (VO_2_ max), is a good indicator of cardiopulmonary health and predictor of cardiovascular mortality. Our aim was to determine the effect of birth weight on aerobic exercise capacity and physical activity. We systematically identified studies reporting exercise capacity (VO_2_ max and VO_2_ peak) and physical activity levels in participants born at VLBW aged eighteen years or older compared to term-born controls from six databases (MEDLINE, OVID, EMBASE, CI NAHL, CENTRAL, and Google Scholar). Meta-analysis of eligible studies was conducted using a random effect model. We screened 6202 articles and identified 15 relevant studies, 10 of which were eligible for meta-analysis. VLBW participants had a lower VO_2_ max compared to their term counterparts (−3.35, 95% CI: −5.23 to −1.47, *p* = 0.0005), as did VLBW adults who had developed bronchopulmonary dysplasia (−6.08, 95% CI −11.26 to −0.90, *p* = 0.02). Five of nine studies reported significantly reduced self-reported physical activity levels. Our systematic review and meta-analysis demonstrated reduced maximal aerobic exercise capacity in adults born at VLBW compared to term-born controls.

## 1. Introduction

According to the World Health Organization (WHO), 15–20% infants worldwide are born at a low birth weight (LBW, <2500 g) [1]. Since the introduction of neonatal intensive care units, there has been a dramatic improvement in survival rates of very low birth weight infants (VLBW, <1500 g) [2,3]. The increased survival has resulted in a focus on morbidity and mortality of these cohorts in later life.

The impact of preterm birth on multi-organ development can have deleterious effects on the cardiopulmonary system [4,5,6,7]. Cardiac magnetic resonance imaging (CMR) has demonstrated structural myocardial changes in adolescents and adults born prematurely which may be associated with reduced functional reserve [4,8]. A study of 102 adults born prematurely demonstrated they had greater left ventricular (LV) mass, smaller internal diameters, and poorer LV strain compared to term-born controls [4]. Mohamed et al. supported those findings reporting smaller LV volumes and reduced LV function in 200 preterm adults [9]. On stress echocardiography, a lower ejection fraction (EF) proportional to exercise intensity was demonstrated [8].

Given the increased prevalence of cardiovascular risk factors described in prematurely born individuals, it is not surprising that birthweight is inversely proportional to adult morbidity and mortality from cardiovascular disease [6,10,11]. In 1991, Barker et al. reported diminished airway function in adults born of reduced birthweight and speculated that this may be secondary to poor prenatal nutrition [12]. Subsequent evidence has shown that LBW is associated with excess respiratory morbidity, independent of or secondary to premature birth or in utero growth retardation (small for gestational age—SGA) [7]. Indeed, prematurity and being born SGA are associated with different risk factors, but many prematurely born infants are SGA. Individuals who developed bronchopulmonary dysplasia (BPD) are particularly at increased risk of chronic respiratory morbidity including an increased requirement for supplementary oxygen following neonatal discharge, more hospital readmissions particularly for respiratory viral infections, and lung function abnormalities persisting even into adulthood. Sadly, such infants may also suffer hearing and visual impairment, feeding difficulties, growth restriction, and chronic kidney disease [13,14].

The impact of preterm birth on multi-organ development can have deleterious effects on the cardiopulmonary system [4,5,6,7]. Furthermore, birthweight has been shown to be inversely proportional to adult morbidity and mortality from cardiovascular disease [8,9].

Measurement of the maximal oxygen consumption (VO_2_ max) is considered the gold standard assessment of cardiorespiratory fitness [15]. It has been shown to be a strong predictor of cardiovascular health, morbidity, and mortality [16,17]. A systematic review of studies of maximal aerobic exercise capacity found a 13% reduction in VO_2_ max in children and adults born prematurely, compared to their term-born counterparts. While the pooled results were significantly different, it was highlighted that the majority of included observational studies showed no significant difference in VO_2_ max between the two groups [18,19,20]. To our knowledge, no systematic review has exclusively focused on determining whether there was an association between VLBW and maximal aerobic exercise capacity in adults.

As a strong predictor of cardiovascular health, an improvement in VO_2_ max may reduce the risk of cardiovascular disease and associated mortality [21]. It is well-established that regular exercise is an effective means of increasing VO_2_ max [22]. In addition to its important association with VO_2_ max, physical activity levels are an important independent protective factor for cardiovascular health [23]. Several studies have suggested that prematurely born individuals are less physically active than their term-born peers, independent of whether they had developed bronchopulmonary dysplasia (BPD) and socio-economic confounders during childhood [24,25,26]. However, the evidence is conflicting [27]; the Epicure study did not reveal any significant differences in physical activity levels between school-age children born prematurely and term-born controls when measured using accelerometers. Therefore, the relationship between birth weight and physical activity levels in adulthood would benefit from further clarification.

The evidence, however, is conflicting [19]. A study of 61 children found no significant differences in physical activity levels assessed using an accelerometer [19]. In addition to its important association with VO_2_ max, physical activity levels are also an independent risk factor that may contribute to cardiovascular morbidity and mortality.

Our primary aim was to undertake a systematic review to evaluate the impact of VLBW on exercise capacity in adults as assessed by VO_2_ max. Our secondary outcome was to compare self-reported physical activity levels between VLBW and term-born adults and assess whether this impacted on exercise capacity.

## 2. Materials and Methods

### 2.1. Methods

This systematic review and meta-analysis was prospectively registered on PROSPERO at https://www.crd.york.ac.uk/prospero/ (accessed on 25 May 2023) as CRD42023429309 [28].

The literature search was conducted according to the Meta-analysis of Observational Studies in Epidemiology (MOOSE) guidelines [29]. The Preferred Reporting Items for Systematic Reviews and Meta-analyses (PRIMSA) guidelines were used to prepare the manuscript [30].

### 2.2. Search Strategy

Relevant studies were identified through searching six electronic databases (EMBASE, OVID MEDLINE(R.), Scopus, CENTRAL, CINAHL, and Google Scholar) between the 1 March 2023 and 15 March 2023. A repeat search was conducted on the 1 June 2023 to identify any further articles that met inclusion criteria. We also hand-searched references from included articles. Search strategies were based on the Cochrane library Neonatal Search Terms [31].

### 2.3. Eligibility Criteria

Studies on exercise capacity in adults (defined as greater than 18 years old) born at VLBW compared to term controls with results of VO_2_ max (mL/kg/min) or VO_2_ Peak (mL/kg/min) using a treadmill or cycle ergometer were eligible. VO_2_ max refers to the maximum rate of oxygen consumption attainable during physical exertion. VO_2_ peak, directly reflective of VO_2_ max, is the highest value VO_2_ attained upon incremental or other high-intensity exercise testing [32]. The recruitment of subjects in some papers was based on gestational age, but only those which reported birthweight were included in the review. BPD was defined as either dependence on supplementary oxygen at 28 days of life or dependence on supplementary oxygen at 36 weeks postmenstrual age (PMA). Other measures of cardiorespiratory exercise capacity were reviewed (such as anaerobic threshold and minute ventilation), but there were insufficient data for a meta-analysis. Given the authors’ capabilities, studies were restricted to those reported in the English language.

Using pre-agreed inclusion criteria, two independent authors (GP and CH) removed duplicates, screened titles and abstracts of retrieved articles, and obtained full-text articles. Any disagreements were resolved through discussion between the two reviewers until a consensus was achieved.

### 2.4. Data Analysis

Data extraction was performed by a single reviewer (GP) using a pre-specified data extraction form. A second reviewer (CH) independently checked the accuracy of the first extraction. Study characteristics, sample size, the method of assessing exercise capacity, and reference values were summarised for each study. For each study, VO_2_ max, VO_2_ peak, and activity levels were extracted for adults born at VLBW and term-born controls.

To be eligible for meta-analysis, a study had to fulfil the following criteria, defined a priori: an original report on the relation between exercise capacity in adults that were born at VLBW, odds ratios (OR), and 95% confidence intervals (95% CI) for exercise tolerance in at least two strata of birth weight. To assess exercise capacity, we analysed results for VO_2_ max and VO_2_ peak. Meta-analyses were conducted using Review Manager (RevMan) version 5.4 [33].

### 2.5. Quality Assessment

The risk of bias for each study was assessed by two independent reviewers (GP and CH) using the Newcastle–Ottawa Scale for cohort and cross-sectional studies [34,35]. Studies were scored across three domains: case selection, comparability, and outcome. Scores across three domains were tabulated to give an overall rating of good, fair, or poor quality. The data extraction for quality was performed by a single reviewer (GP) and three randomly chosen papers were checked for consistency by a second reviewer (CH), with no discrepancies being identified. For cohort studies, we considered that participants lost to follow-up were unlikely to introduce bias if follow-up rates were greater than 80% or between 70% and 80% with an accompanying statement describing those lost to follow up.

## 3. Results

### 3.1. Identified Studies and Characteristics

The course of the systematic review is outlined in a PRIMSA 2020 flow diagram (Figure 1). Seven thousand, eight hundred and seven studies were identified through database searching. A total of 1605 duplicates were removed, and 6202 abstracts were screened. Eighty-one full-text articles were screened for eligibility, and the quality of fifteen studies was evaluated [8,24,36,37,38,39,40,41,42,43,44,45,46,47]. The characteristics of the studies included are summarised in Table 1. From all included studies, there were 1132 VLBW participants, 914 controls, and 75 VLBW who had had BPD. Individuals were born between 1984 and 1998. Participants were assessed at ages 18 to 30 years old [8,24,36,37,38,39,40,41,42,43,44,45,46,47].

### 3.2. VO_2_ Max in VLBW Infants

Ten studies assessed VO_2_ max in VLBW adults compared to term-born controls [24,36,37,40,41,43,44].

Three studies undertook a subgroup analysis for VLBW individuals who had BPD, including 75 participants. Table 2 summarises the characteristics and results of the studies included in the analysis. The average weight of VLBW adults at follow-up was 68.85 kg, compared to 73.75 kg in the control group (Student’s *t* test, *p* = 0.034). Four studies assessed VO_2_ max, three of which used a cycle ergometer [8,36,37,46]. Six studies assessed VO_2_ peak using a combination of cycle ergometry and treadmill exercise protocols [24,40,41,42,43,47].

Meta-analysis indicated that adults born at VLBW had a significantly lower VO_2_ max/VO_2_ peak compared to controls (mean difference: −3.35 [95% CI −5.23 to −1.47] mL/kg/min, *p* = 0.0005) (Figure 2). However, there was high heterogeneity between studies included in the analysis (I^2^ = 87%). Meta-analysis of studies solely reporting on those who had BPD found they were more likely to have a lower VO_2_ max (mean difference: −6.08 [95% CI −11.26 to −0.90] mL/kg/min, *p* = 0.02,) than term-born controls. There were no significant differences in VO_2_ max between VLBW participants who had BPD and those that did not (*p* = 0.33).

### 3.3. Levels of Physical Activity in VLBW Infants

Nine studies from five different countries reported on physical activity levels in VLBW adults compared to controls (Table 3) [8,26,37,38,39,42,44,47]. Three studies followed up individuals from the Helsinki Study of VLBW Adults [39,41,44]. Three studies used the European Community Respiratory Health Survey II to identify adult’s physical activity levels [36,44,47,51]. Five studies did not report on questionnaires or tools utilised to assess physical activity levels [8,24,37,39,42].

Five studies found significant differences in self-reported activity levels in VLBW adults compared to controls [36,39,44,45,47]. In four studies, VLBW adults were less likely to engage in weekly vigorous physical activity [36,44,45,47]. In one study, despite no significant difference in the frequency of exercise, VLBW adults were more likely to engage in less intense physical activity for a shorter duration of time [39]. Four studies found no significant difference in the frequency of physical activity between VLBW and control groups [8,24,37,42]. Due to a difference in measurable outcomes and assessment tools, the results were unsuitable for meta-analysis.

## 4. Discussion

We have demonstrated that exercise capacity is significantly reduced in adults born at VLBW, independent of whether they had BPD, compared to term born controls. It is important to consider the origin of differences in VO_2_ max between VLBW adults and TB term-born controls. Maximal aerobic exercise capacity is impacted by age, sex, weight, size, body composition, and physical activity levels. Due to a lack of data reported in individual papers, we were unable to analyse results to determine if there were differences related to sex. A follow-up study of 150 adults born prematurely recruited into the United Kingdom Oscillation Study (UKOS) found males compared to females completed significantly greater distances during shuttle sprint testing and reported exercising more each week [52]. While VO_2_ max was not assessed in that study, sex differences in the amount of exercise undertaken could potentially impact on VO_2_ max. Furthermore, in a study of elite endurance athletes, women were found to have a VO_2_ max 10% lower than their male counterparts [53].

Given the absolute value is highly impacted by body weight, VO_2_ max and VO_2_ peak are typically expressed as milliliter/kg/minute. While this enables results to be adjusted for body weight, body composition remains a likely confounder. Eight out of ten included studies reported the participants’ weight [8,24,36,37,40,41,43,46,47]. On pooled analysis, there was a significant difference in the mean weight of adults born at VLBW compared to term-born controls. While impossible to predict based exclusively on weight, BMI is generally well-correlated to percentage body fat [54,55,56]. One study of 25 female athletes aged between 17 and 22 years found a non-significant negative correlation between percentage body fat and VO_2_ max [57]. Goren et al. demonstrated a strong correlation between fat-free mass (FFM) and VO_2_ max [58]. This may explain the greater effect size observed in our meta-analysis which focused exclusively on adults compared to the results of Edwards et al. which also included children [18]. Dual energy X-ray absorptiometry (DXA) of 433 healthy subjects demonstrated a decline in FFM with age [59]. This emphasises the importance of including anthropometric measurements in studies investigating maximal aerobic exercise capacity in future.

There is a well-established association between physical activity levels and maximal aerobic exercise capacity. Meta-regression and analysis of 28 articles highlighted an increase in VO_2_ max with physical activity training, independent of the volume and intensity of exercise sessions [60]. Crowley et al. reported in their systematic review that both high and low intensity training when undertaken frequently increased VO_2_ max [61]. Interestingly, despite adjusting for physical activity levels, Gostelow and Stohr found a significantly lower VO_2_ max in individuals born at VLBW [19]. In our review, five studies reported that adults born at VLBW exercised less than their term-born counterparts [36,39,44,45,47], whereas four studies found no such association [4,8,24,38]. This highlights the need for further research assessing the relationship of physical activity levels and VO_2_ max in relation to birth weight. If frequent exercise improves maximal aerobic exercise capacity, an important predictor of cardiovascular morbidity and mortality, a pertinent public health strategy would be to target educational interventions and physical activity programs at VLBW adults.

Interestingly, despite adjusting for PA levels, Gostelow and Stohr found a significantly lower VO_2_ max in individuals born at VLBW [52,53]. There is improvement in maximal aerobic exercise capacity with regular exercise, that PA may be used as a potential intervention [52,53]. This raises a question as to whether VLBW infants should be recommended targeted exercise regimens as a preventative cardiovascular strategy. The physiological mechanisms resulting in reduced maximal aerobic exercise capacity in adults born at VLBW remain poorly understood. Aerobic exercise capacity is determined by the integrative responses of the cardiovascular and respiratory systems, in addition to oxygen uptake by skeletal muscles [62]. Several studies have demonstrated that [54]. Our findings support previous research demonstrating a reduced maximal aerobic exercise capacity, independent of prematurity-related perinatal factors such as BPD [40]. Pulmonary gas exchange during exercise, assessed by the alveolar-to-arterial oxygen difference (A-aDO_2_), is comparable between prematurely born and term-born TB adults during exercise [40,43,63]. It has however been hypothesised that adults born at VLBW may have higher airway resistance and smaller peripheral airways, requiring a greater concentration of oxygen to maintain ventilation respiration during exercise [64]. Follow-up of the UKOS cohort found males born prematurely were more likely to have poorer smaller airway function, however, they performed better on exercise testing compared to their female counterparts, possibly indicating that other factors’ physiological mechanisms may have a greater influence [52]. Adults born prematurely have been shown to have a significantly increased pulmonary arterial pressure during exercise, which may reduce pulmonary blood flow and subsequently VO_2_ max [64,65]. Physiological mechanisms contributing to a reduced maximal aerobic exercise capacity are multi-factorial and complex, where further research is required to fully understand the impact of birth weight and prematurity.

In addition to a reduced VO_2_ max, a predictor of increased cardiopulmonary mortality, adults born at VLBW are also at higher risk due to the increased prevalence of hypertension [66], heart failure [67], diabetes [68], and cardiometabolic syndromes [66]. Given this, a more detailed cardiovascular risk assessment in adults known to be born at VLBW may be of benefit. One suggestion is to screen adults born at VLBW in general practice using a risk scoring system, such as the widely utilised QRISK2, to predict individuals 10-year risk of cardiovascular disease [69]. It would be important however to evaluate the financial cost, resource implications, and the most appropriate and effective age-range to target such an intervention.

Furthermore, adults born preterm have been shown to have a significant increase in pulmonary arterial pressure during exercise, which may reduce pulmonary blood flow and subsequently VO_2_ max [59,60]. Physiological mechanisms contributing to the observed reduction in maximal aerobic exercise capacity are multi-factorial and it is clear further research is required to fully understand the impact of birth weight and prematurity.

Adults born at VLBW are at a higher risk of hypertension [61], heart failure [62], diabetes [63], and cardiometabolic syndromes [61]. Interestingly, despite a greater prevalence of cardiovascular risk factors, an association with ischemic heart disease remains inconclusive [64,65]. An increased relative-risk of all-cause mortality in adults born prematurely, however, is well-established [66]. A more detailed cardiovascular health assessment in adults known to be born at VLBW may be of benefit, but evaluation of the efficacy and resource implications of adult-targeted interventions would be important. Studies such as the trial of exercise to prevent hypertension in young adults are therefore very welcome, even though only 38.7% of those born prematurely were born at less than 32 weeks of gestation [70].

Despite our efforts to generate a precise effect of being born at VLBW on maximal aerobic exercise capacity, our review has some limitations. On analysis, there was a high proportion of heterogeneity between studies reporting maximal aerobic exercise capacity. In part, this may be secondary to our decision to include studies reporting VO_2_ peak and VO_2_ max, however, prior studies have shown that VO_2_ peak is reflective of VO_2_ max [32]. The heterogeneity is possibly attributable to different methodologies and protocols used between studies to estimate maximal aerobic exercise capacity. Eight studies utilised cycle ergometry [8,36,37,39,41,44,46,47] whereas two studies utilised a treadmill [24,27]. While studies utilising a treadmill demonstrated comparable results to those using a cycle ergometer in this review, prior studies have commented on lower values of VO_2_ max using cycle ergometry when intra-subject comparisons of both methods were utilised in the same study [71,72]. While challenging, this perhaps highlights a need to standardise methodology and protocols utilised to measure maximal aerobic exercise capacity.

All participants included in the meta-analysis were between their second and third decade of life, a period well-established to correlate to peak maximal aerobic exercise capacity. Generally, it is estimated that VO_2_ max declines 10% per decade after the age of 25 years and 15% between the ages of 50 and 75 [73,74,75]. Most studies included in our meta-analysis followed up participants in their third decade of life, with latter follow-up. It will be interesting to observe the impact of age on differences in VO_2_ max between adults born at VLBW and at term.

Due to differences in outcome measures between studies, a meta-analysis could not be performed to assess self-reported physical activity levels in adults born at VLBW. Given the correlation between physical activity levels and cardiovascular morbidity and mortality, in addition to the possibility of reduced activity levels in adults born at VLBW, standardisation of outcome measures between studies is of vital importance. In studies evaluating maximal aerobic exercise capacity and physical activity levels, it is important to critically evaluate participant recruitment given the high risk of recruitment bias associated with exercise-based studies [76].

## 5. Conclusions

In conclusion, maximal aerobic exercise capacity was significantly reduced in adults born at VLBW compared to term-born controls. Given the relationship between exercise capacity and cardiovascular morbidity and mortality, this could have significant implications for individuals’ long-term health. The variability in outcome measures assessing physical activity meant it was difficult to accurately assess the association with birthweight. We recommend a standardised approach of assessing physical activity levels for future studies, such as the European Respiratory Health Community Questionnaire II or Metabolic Equivalent of a Task levels.

## Figures and Tables

**Figure 1 children-10-01427-f001:**
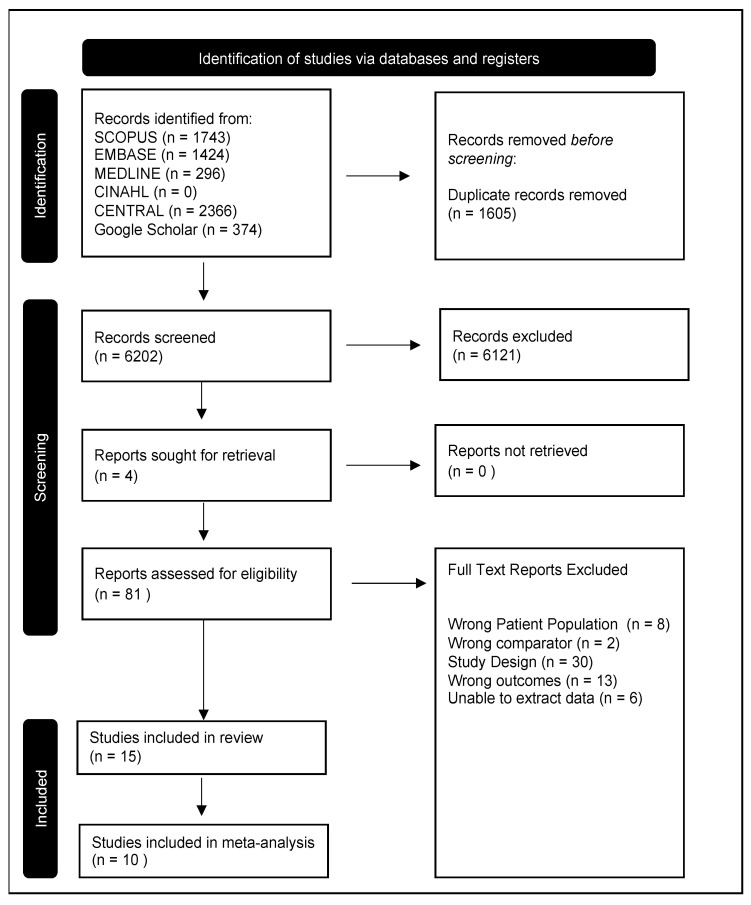
Preferred Reporting Items for Systematic Reviews and Meta-analyses (PRIMSA) flow diagram outlining the course of our systematic literature search for articles evaluating maximal aerobic exercise capacity in adults born at very low birth weight. Six databases (SCOPUS, EMBASE, MEDLINE, CINAHL, CENTRAL, and Google Scholar) were searched. A total of 6202 articles were screened and 81 articles assessed for eligibility. Fifteen studies were included in the review and ten in the meta-analysis.

**Figure 2 children-10-01427-f002:**
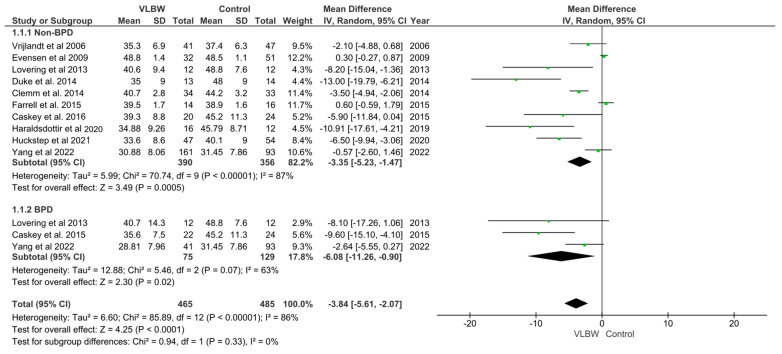
Forest plot of aerobic exercise capacity (VO_2_ max/VO_2_ peak) in adults born at very low birth weight compared to their term-born counterparts [24,36,37,40,43,44,46,47,48,50].

**Table 1 children-10-01427-t001:** Study characteristics of included studies.

Author	Country	Study Design	Number of VLBW Infants	Number of Control Subjects	Age at Follow Up (Years)	Outcome Measures
Vrijlandt et al., 2006 [36]	Netherlands	Prospective cohort study	42	48	18–22	VO_2_ max Physical activity level
Evensen et al., 2009 [37]	Norway	Prospective cohort study	32	51	18	VO_2_ max
Narang et al., 2009 [38]	UK	Prospective cohort study	57	50	20–25	Physical activity level
Sipola-Leppanen et al., 2011 [39]	Finland	Prospective cohort study	116	118	20–28	Physical activity level
Lovering et al., 2013 [40]	USA	Prospective cohort study	12	12	18–27	VO_2_ peak
Clemm et al., 2014 [24]	Norway	Prospective cohort study	34	33	24–25	VO_2_ peak VO_2_ peak at anaerobic threshold Physical activity level
Duke et al., 2014 [48]	USA	Prospective cohort study	13	14	20–25	VO_2_ peak
Saarenpaa et al., 2015 [42]	Finland	Prospective cohort study	160	162	20–25	Physical activity level
Farrell et al., 2015 [43]	USA	Prospective cohort study	14	16	20–23	VO_2_ max
Caskey et al., 2016 [44]	UK	Prospective cohort study	20	24	23–30	VO_2_ peak
Kasaeva et al., 2012 [45]	Finland	Prospective cohort study	94	101	21–27	Physical activity level
Haraldsdottir et al., 2020 [46]	USA	Prospective cohort study	12	12	24–28	VO_2_ max at normoxia and hypoxia
Huckstep et al., 2018 [8]	UK	Prospective cohort study	47	54	20–26	VO_2_ max Physical activity level
Yang et al., 2022 [47]	New Zealand	Prospective cohort study	202	93	26–30	VO_2_ peak Physical activity level
Cheong et al., 2023 [49]	Australia	Prospective cohort study	128	126	25	Six-minute walk test Maximum beep test level

**Table 2 children-10-01427-t002:** Summary of physical maximum aerobic exercise capacity in adults born at very low birth weight.

Study	Birth Weight VLBW Infants (g)	Birth Weight Control Subjects (g)	Age at Follow Up (years)	Weight VLBW Adults (kg)	Weight Control Group (kg)	VO_2_ Max/Peak Measurement	VO_2_ Measurement in VLBW Group (mL/kg/min)	VO_2_ Measurement in Control Group (mL/kg/min)	VO_2_ Measurement in BPD Group (mL/kg/min)
Vrijlandt et al., 2006 [36]	1246 ± 232	-	18–22	65 ± 10	72 ± 10	VO_2_ Max Cycle Ergometer	35.3 ± 6.9	20.8 ± 1.2	-
Evensen et al., 2009 [37]	1245 (800–1500)	3700 (2670–5140)	18	64.2 ± 1.7	69.8 ± 1.3	VO_2_ Max Treadmill	48.8 ± 1.4	48.5 ± 1.1	-
Lovering et al., 2013 [40]	1160 ± 450	-	21–24	64.7 ± 9.3	75.7 ± 10.4	VO_2_ Peak Cycle Ergometer	40.6 ± 9.4	48.8 ± 7.6	40.7 ± 14.3
Clemm et al., 2014 [24]	1173 ± 163	-	24–25	71.5 ± 4.3	72.3 ± 5.9	VO_2_ Peak Treadmill	40.7 ± 2.8	44.2 ± 3.2	-
Duke et al., 2014 [48]	1080 ± 430	-	20–25	65 ± 10	72 ± 12	VO_2_ Peak Cycle Ergometer	35.0 ± 9.0	48.0 ± 9.0	-
Farrell et al., 2015 [43]	1027 ± 296	>1500	20–23	76.3 ± 5.0	71.8 ± 5.4	VO_2_ Peak Cycle Ergometer	39.5 ± 1.7	38.9 ± 1.6	-
Caskey et al., 2016 [44]	1234 ± 205	3569 ± 297	21–30	-	-	VO_2_ Peak Cycle Ergometer	45.2 ±11.3	39.3 ± 8.8	35.6 ± 7.5
Haraldsdottir et al., 2020 [46]	<1500 g	-	24–28	70.1 ± 13.3	75.6 ± 0.7	VO_2_ Max Cycle Ergometer	34.88 ± 9.26	45.79 + 8.71	-
Huckstep et al., 2021 [50]	1916 ± 806	3390 ± 424	22–27	-	-	VO_2_ Max Cycle Ergometer	33.6 ± 8.6	40.1 ± 9.0	-
Yang et al., 2022 [47]	1131 ± 233	3362 ± 529	28–29	74.1 ± 18.8	80.8 ± 16.3	VO_2_ Peak Cycle Ergometer	30.46 ± 8.06	31.45 ± 7.86	28.3 ± 1.1

**Table 3 children-10-01427-t003:** Summary of physical activity levels in adults born at very low birth weight.

Study	WeightVLBWInfants (g)	WeightControl Subjects (g)	Age at Follow UpVLBW Infants(Years)	Age at Follow Up Control Subjects(Years)	PA Assessment Measure	Results	Summary of Impact
Vrijlandt et al., 2006 [36]	1246 ± 232	-	19 ± 0.3	20.8 ± 1.2	European Community Respiratory Health Survey IIMean hours of vigorous exercise per week	Preterm born infants undertake significantly less vigorous exercise per week (1.9 h ± 2) compared to term born controls (2.9 h ± 2)	
Narang et al., 2009 [38]	1440 ± 550	3410 ± 2390	21.7 ± 1.2	23.1 ± 2.0	No formal questionnaire reportedMean days engaged with physical activity per week	There was no statistical difference in time spent being active per week between VLBW infants (3.0 ± 2.42) and the control group (3.0 ± 1.79)	
Sippola-Leppanen et al., 2011 [39]	1125 ± 223	3606 ± 469	22.3 ± 2.2	22.6 ± 2.2	No formal questionnaire reportedFrequency, duration, and intensity of exercise	There was significant difference between VLBW and control subjects in the intensity and duration of physical activity during a typical weekThere was no difference in the frequency of activity between groups	
Kaseva et al., 2012 [45]	1157 ± 208.7	3608 ± 492	24.9 ± 2.1	25.1 ± 2.2	Modified Kuopio Ischaemic Heart Disease Risk Factor StudyFrequency, time, and intensity of conditioning exerciseFrequency, time, and intensity of leisure-time physical activity	No significant difference in commuting, leisure-time, or conditioning physical activity between groupsVLBW infants were more likely to do less vigorous activity compared to their counterparts	
Clemm et al., 2014 [24]	1173 ± 163	-	24.7 ± 1.2	25.1 ± 1.2	No formal questionnaire reportedCategorical hours spent exercising per week	No statistically significant difference in leisure time spent doing physical activity between EP and term-born individuals.	
Caskey et al., 2016 [44]	1234 ± 205	3569 ± 297	26.4 ± 3.7	28.3 ± 3.3	European Community Respiratory Health Survey IIFrequency exercised 2–3 h per week	Statistically significant difference in the frequency individuals exercised 2–3 h per week between BPD, non-BPD adults, and term-born controls	
Saarenpaa et al., 2015 [42]	1126 ± 218	3599 ± 466	22.4 ± 2.1	22.5 ± 2.5	No formal questionnaire reportedFrequency of exercise per week	No statistically significant difference in the frequency of exercise per week between VLBW and non VLBW individuals	
Huckstep et al., 2018 [8]	1916 ±806	3390 ± 424	22.7 ± 3.04	23.6 ± 3.8	No formal questionnaire reportedHours spent doing moderate and vigorous physical activity per week	No statistical difference in hours spent doing moderate or vigorous activity between groups	
Yang et al., 2022 [47]	1131 ±233	3362 ± 529	28.3 ± 1.1	28.2 ± 0.9	European Community Respiratory Health Survey IIMean days engaged with physical activity per week	VLBW exercised significantly less (2.9 h ± 2.6) per week compared to term-born controls (37 ± 2.4).

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
