# Peer review of "Exercise Capacity in Very Low Birth Weight Adults: A Systematic Review and Meta-Analysis"

_children, 2023, doi:10.3390/children10081427_

Round 1

Reviewer 1 Report

Introduction- can the authors provide more details regarding the impact of preterm birth on the later development of the cardiovascular system?

Results: Is there a possibility to evaluate the effect of gender on exercise capacity?

Discussion:

Can the authors discuss the multi-organ effects of prematurity/VLBW in parallel to the cardiovascular system (respiratory, metabolic, renal, musculoskeletal..)?

The authors may want to mention specific follow-up programs / interventions for adults born VLBW/preterm in order to improve quality of life and health (in addition to standardize evaluation of physical activity). 

Reviewer 2 Report

This manuscript about exercise capacity in VLBW at adult age is a pertinent work. 

Some part of this paper can be improved:

Please insist if VO2 max is a marker or result from pulmonary development and physical activity, so that physical activity of VLBW during childhood could be in the spotlight.

As you introduce is the discussion section, preterm and small for gestational age are two distinct situations, please insist on it.

Please check for your acronyms (PRISMA and MOOSE)

Do not limit your flow chart to english langage study, and clarify your flow chart figure.

manuscript have to be improved to be accepted
